# Let’s Talk about Sex… and ADHD: Findings from an Anonymous Online Survey

**DOI:** 10.3390/ijerph20032037

**Published:** 2023-01-22

**Authors:** Susan Young, Larry J. Klassen, Shayne D. Reitmeier, Jake D. Matheson, Gisli H. Gudjonsson

**Affiliations:** 1Psychology Services Ltd., London CR9 7AE, UK; 2Department of Psychology, University of Reykjavik, 110 Reykjavik, Iceland; 3Eden Mental Health Centre, 1500 Pembina Avenue, Winkler, MB R6W 1T4, Canada; 4Department of Family Medicine, Rady Faculty of Health Sciences, Max Rady College of Medicine, University of Manitoba, Winnipeg, MB R33 0W2, Canada; 5ReitMD, Portage la Prairie, MB R1N0S8, Canada; 6Department of Psychology, Institute of Psychiatry, Psychology & Neuroscience, King’s College London, London SE5 8AF, UK

**Keywords:** attention-deficit/hyperactivity disorder (ADHD), sex, psychosexual, risk

## Abstract

Attention-deficit/hyperactivity disorder (ADHD) is a neurodevelopmental disorder characterized by inattention, hyperactivity and impulsivity. A dearth of studies have investigated psychosexuality in this population, often with few (predominantly male) participants. We recruited individuals with and without ADHD via an anonymous online survey distributed electronically by ADHD support organizations and social media. We investigated sexual history; interests and practices; and relationships. Of 1392 respondents, we classified an ‘ADHD’ group (*n* = 541; 30.5% male) and compared them to individuals of similar ages without ADHD, (‘Other’ group; *n* = 851; 37.6% male). The ADHD group (both males and females) had a significantly higher preference for same-sex or either-sex partners; and higher rates of electronic sexual exchanges, masturbation, and sexually transmitted diseases. They were more adventurous in sexual interests and practices and substantially less satisfied with their partners, both sexually and generally. Within the ADHD group, significant sex differences emerged: females had younger onset of sexual activities, used contraception less frequently, had more sexual partners and practiced more infidelity. Sexual interests differed between the sexes, but females more commonly acted on them, whereas males did not. Findings suggest both sexes engage in risky sexual behaviors, perhaps driven by impulsivity, but risk is substantially greater for females with ADHD.

## 1. Introduction

Attention-deficit/hyperactivity disorder (ADHD) is a childhood-onset neurodevelopmental disorder characterized by pervasive behavioral symptoms of hyperactivity, inattentiveness and impulsivity that have been present for at least 6 months and adversely impact daily functioning and development [1]. It is associated with impaired functioning that impacts daily functioning and development [2]. Research has focused on health, social, educational, occupational and community outcomes but less so on psychosexual outcomes. Psychosexual functioning refers to the mental, emotional and behavioral aspects of sexual development and is recognized to be important for individuals’ overall health status and quality of life. In addition to a reduced focus in research on this topic, from a clinical perspective, the area of psychosocial functioning is often overlooked or only minimally discussed, missing an important opportunity to provide psychoeducation and potentially help reduce the risk for adverse outcomes.

Early studies noted an association between poor relationship satisfaction, a high incidence of divorce among participants with diagnosed ADHD and risky sexual behavior in young adulthood [3,4,5]. Compared with community controls (i.e., normal adults from the local community), the Milwaukee Group follow-up of young people with hyperactivity at age 21 and 27 years [6,7] reported participants engaged in sexual intercourse at a younger age, had a greater number of sexual partners and were at increased risk of early pregnancy and sexually transmitted infections (STIs). Risky sexual behavior continued into adulthood for individuals who remained symptomatic. The hyperactive group continued to practice riskier sexual behavior than the community control group, suggesting that remission in symptoms may reduce the extent of riskier sexual behavior [7].

More recent studies comparing young people and adults with ADHD with typically developing controls have found a range of poor psychosexual outcomes for those with ADHD [8,9,10]. These include relationship dissatisfaction; risky sexual behaviors (RSBs) and subsequent outcomes (e.g., higher incidences of STI’s and unwanted pregnancies); sexual dysfunctions and atypical sexual disorders, such as hypersexuality and paraphilia [9,10].

In a recent systematic literature review, Soldati et al. [11] found that participants with ADHD reported more sexual interest, higher masturbation frequency, more sexual dysfunction and lower sexual satisfaction compared with general population controls. The authors suggest that the elevated level of dysfunction in the ADHD group, both among males and females, creates potential problems in romantic relationships. Citing the work of Bijlenga et al. [12] and Pera [13], Soldati and colleagues suggest that masturbation and other sexual activities may function as coping strategies for individuals with ADHD by releasing tension, regulating their emotions and helping them to cope with stressful situations.

Some studies have compared male and female sexual functioning. Compared with controls, both males and females with ADHD have been found to engage in risky sexual behavior that carries an increased risk of developing STIs [14], to have less satisfaction with their sex lives [12] and to have greater sexual dysfunction [15]. However, Abdel-Hamid et al. [16] found that sexual dysfunction was no longer significant when adjusting for anxiety and depression, suggesting that comorbid symptoms may be an important mediator.

However, most studies into sexual behavior of people with ADHD have focused on males, either exclusively or as the majority proportion of participants. The number of participants is often low [9,10]. Hence, the present study comprised a population both with and without ADHD in an online survey. The sample size was large and included both females and males, which made it possible to study gender differences in sexual history, interests and behavior; and their relationships with ADHD diagnosis and current symptom severity. This method provides the opportunity to obtain anonymous data from a large population, providing the potential for greater reliability when investigating sensitive personal topics of a sexual nature that represent real-life aspirations and experiences, which in turn will be more generalizable.

One of the main advantages of the current survey is that we measured both relationship satisfaction and sexual satisfaction, and importantly explored the association between them within the context of ADHD symptom severity and other potentially confounding variables. To the best of our knowledge, this has not been done before, and it adds a novel approach to the analysis of the data.

Following the methodology that Vowels and Mark [17] recommend for studying the association between relationship satisfaction and sexual satisfaction, we explore the relative contributions of age, sex (female/male), severity of ADHD symptoms, currently being on prescribed ADHD medication, history of anxiety/depression diagnosis, and general relationship satisfaction on predicting overall sexual satisfaction as an outcome measure. 

The research questions were as follows:Is ADHD associated with sexual orientation; sexual interests and practices; and riskier sexual behavior?Are there differences between ADHD and other participants in the motivation behind extra-partner affairs?Does an ADHD diagnosis reduce the association between relationship satisfaction and sexual satisfaction in one’s current relationship?To what extent do ADHD symptom severity and associated clinical conditions adversely affect the extent of sexual satisfaction in relationships?

## 2. Materials and Methods

### 2.1. Participants

There was a total of 1466 participants from Canada (*n* = 619), the United Kingdom (*n* = 556), USA (*n* = 126), Denmark (*n* = 40), Turkey (*n* = 22) and other countries (*n* = 103). Nearly all participants specified their gender (1456; 93.3%), classifying themselves as female (907; 62.3%), male (485; 33.3%) and non-binary (64; 4.4%). Seventy-five (5.2%) participants responded that their current gender did not match their sex assigned at birth, the majority of whom (54; 72%) were in the non-binary group. All of the (non-binary) participants were excluded from the current analyses and will be presented in a separate article. This left a sample of 1392 participants, of whom 1368 (98.3%) gave their age. The mean ages for females and males were 38.1 (SD = 11.9; range 18–76 years) and 41.4 (SD = 13.8; range 18–78 years), respectively. Males were older (low effect size;(t = 4.6, *p* < 0.001; Cohen’s d = −0.262).

### 2.2. Measures

A 34-item questionnaire was developed for the survey consisting of four separate parts (available by contacting the lead author). There was no corroborative information collected to support the answers from the questionnaire for pragmatic reasons and issues of confidentiality (i.e., this was an anonymous online survey).

#### 2.2.1. Part 1 (‘Basic Information’)

This asked for background information (i.e., current age, country, gender, sexual preferences, clinical diagnosis and ADHD medication). Participants were not asked about other medications, since the primary focus of the study was on ADHD.

For clinical diagnosis, the participants were asked from a pre-determined list ‘Has a health practitioner ever diagnosed you with any of the following conditions (check all those that apply):’ (see Table 1 for list of symptoms contained within the questionnaire).

This section also contained the six-item Adult ADHD Self-Report Scale [ASRS] [18], rated from 0 (‘Never’) to 4 (‘Very often’). Scores range from 0 to 24.

#### 2.2.2. Part 2 (‘Sexuality in Relationships’)

This focused on sexual activity and infidelity across one’s lifespan. For those participants who reported currently being in a romantic relationship, these topics were also explored separately with respect to their current respective partners, together with their satisfaction with their relationships.

‘Sexual activity’ was defined as any interaction that is sexual in nature for the purposes of seeking pleasure. The reader was informed that this included (but was not limited to) oral, anal and vaginal intercourse; genital stimulation; and the use of adult toys regardless of whether orgasm/ejaculation was reached.

A ‘romantic relationship’ was defined as two people who are emotionally close. The reader was informed that terms often used to describe this type of relationship are boyfriend, girlfriend, partner and/or spouse. Casual sexual encounters or ‘friends with benefits’ were not included in this definition.

Some questions asked about behavior ‘in a typical month’. These questions referred to experiences that occurred before the COVID-19 pandemic when social distancing measures were not in force.

#### 2.2.3. Part 3 (‘Sexual History’)

This included items about age of the participant when they first engaged in consensual sexual activity, number of sexual partners over their lifetime, extra-partner sexual engagement whilst in a committed relationship, factors that motivated the participant to engage in extra-partner relationships, use of contraception and STIs.

#### 2.2.4. Part 4 (‘Sexual Interests and Practices’)

This focused on practices associated with pornography, masturbation, electronic exchanges of a sexual nature and specific sexual interests and consensual activities in which they have engaged.

### 2.3. Procedure

The survey was designed using the ‘SurveyMonkey’ website in the English language. Draft formats of the survey were sent to collaborating ADHD organizations (the ADHD Foundation, Attention Deficit Disorder Information and Support Service (ADDISS) and the Danish ADHD Association) for review and feedback. This led to several revisions. In an introduction to the survey, readers were informed that the purpose of the survey, which took around 20 min to complete, was to better understand the sexual interests and practices of adults with and without ADHD. To be eligible, participants had to be 18 years or older. They were informed that their responses were anonymous and not linked to any personally identifying information. Participation was voluntary, and if they chose to participate, they were informed they had the option of only responding to questions they felt comfortable answering. Other than age (in years and geographical location), there followed 32 questions presented in either a closed option format (i.e., only one of several options could be claimed) or in an open option format (i.e., by checking all options that applied from a drop-down menu). At the end of the survey, participants had the opportunity to add written comments.

Thus, a convenience sample was obtained via this anonymous online survey; links to the survey were shared and distributed electronically by the ADHD organizations and through social media. The survey was open for three months until 12 January 2021, and the data were exported into SPSS for analysis.

### 2.4. Analytical Strategy

Means and standard deviations are provided for continuous variables, including age (*t*-test) and number of lifetime sexual partners (Mann–Whitney U test due to skewed data). Chi-square tests were used for categorical variables. Phi was used to measure effect size for 2 × 2 contingency tables. Cramer’s V was used to measure the respective effect sizes with tables larger than 2 × 2.

We used Cohen’s d [19] to measure effect sizes regarding correlation coefficients: 0.10, 0.30 and 0.50 measured small, medium and large effect sizes, respectively. When measuring the effect sizes of t-tests, we used Cohen’s recommended cut-off scores of 0.20, 0.50 and 0.80 for small, medium and large effect sizes, respectively. All tests were two-tailed.

Hierarchical linear regression models were conducted to investigate whether variables of interest explain a significant amount of variance in dependent variables after accounting for other variables (e.g., age, sex and ADHD symptoms).

Most studies on the association between relationship satisfaction and sexual satisfaction have suggested that the link is bidirectional, but there is evidence from longitudinal research for directionality and that relationship satisfaction is a better predictor of sexual satisfaction than vice versa [17]. We used this directional model to investigate how much of the variance in sexual satisfaction is accounted for by relationship satisfaction after adjusting for age, sex, ADHD symptoms and historical diagnoses of anxiety and depression.

## 3. Results

### 3.1. Part 1: Basic Information

#### 3.1.1. Clinical Conditions

Table 1 shows the range of clinical conditions the participants claimed from a pre-determined list of 18 different conditions. Out of the 1392 participants, 864 (62.1%) claimed one or more clinical condition. The most common conditions were ADHD (38.9%), followed by depression (30.0%) and anxiety disorder (29.0%). There was an overlap between the conditions of anxiety, depression and post-traumatic stress disorder; and these conditions, either individually or combined, were reported by 556 (39.9%) of participants. These three conditions were aggregated into one clinical category for further analysis. Anxiety and depression were the most common co-morbid conditions found among the ADHD group, 211 (39.0%) participants in both groups, respectively. Autism spectrum disorder, Asperger’s syndrome and social communication disorder all overlapped and were combined into one group labelled ‘ASD’ consisting of 66 (4.7%) participants.

Among the 541 ADHD participants, 46 (8.5%) had comorbid ‘ASD’. Of the 66 participants with ‘ASD’, over two thirds (69.7%) also claimed an ADHD diagnosis. For this reason, only the findings for the ADHD group are specifically provided in the current study. The sample of ASD participants without ADHD was too small for any meaningful independent analysis (*n* = 20).

Having made these adjustments, the participants in the two groups were compared for further analysis. These were the ‘ADHD’ group (*n* = 541, 38.9% of the total sample, of whom 30.5% were male) and data from all other participants, labelled the ‘Other’ group (*n* = 851, 61.1% of the total sample, of whom 37.6% were male). There was no significant age difference between the ADHD (38.8 years) and Other (39.6 years) participants. 

#### 3.1.2. ADHD Medication and Severity of ADHD Symptoms

Out of the 1373 participants who completed the ASRS, the mean score among the ADHD group was 15.88 (SD = 4.71; *n* = 538), and it was 9.54 (SD = 4.71; *n* = 835) for the Other participants. This difference between the two groups was significant (t = 26.32, df = 1361, *p* < 0.001; Cohen’s d = 1.45; large effect size).

Both age and sex were related to the severity of symptoms. Age correlated negatively (younger age more symptoms) with total ADHD symptoms (r = −0.175, *p* < 0.01, *n* = 1373, low effect size). Females (12.69, SD = 5.3, *n* = 896) had a significantly higher total symptom score than males (10.76, SD = 5.22, *n* = 477); (t = 6.46, *p* < 0.001, Cohen’s d = 0.366, medium effect size). 

1360 participants responded to the question asking about ADHD medication status, and 286 (21.0%) claimed that they were currently taking prescribed ADHD medication. All but six (2.1%) of those 286 participants reported an ADHD diagnosis. When focusing only on the ADHD-diagnosed group, 280 (51.8%) reported being currently prescribed ADHD medication and regularly taking it. This medication group is included in Table 2. A further 71 ADHD participants (13.1%) reported being prescribed ADHD medication but not taking it daily.

Three hierarchical regression models (see Table 2) were conducted to investigate the relative impacts of age, sex, and anxiety/depression/PTSD diagnoses (Step 1), ADHD diagnosis (Step 2), and current ADHD medication status (Step 3). The final model explained 39.4% of the variance in ADHD symptoms; the single greatest contribution was for ADHD diagnosis, followed by a diagnosis of anxiety/depression/PTSD, young age and being a female. A separate analysis, adding the ASD group to the model, added nothing to the overall variance due to the overlap with the ADHD group, and this is therefore not further reported on.

### 3.2. Part 2: Sexuality in Relationships

#### 3.2.1. Sexual Orientation

Out of 1371 participants who gave a specific gender preference (i.e., ‘female’, ‘male’, ‘any gender’, ‘both females and males’), 974 (71.0%) had a sexual preference for the opposite sex. A further 18 participants (17 females and one male) claimed that none of the categories provided applied.

Table 3 provides the claimed sexual preferences for the ADHD and Other participants, respectively (i.e., “Who do you find to be sexually attractive?” from the pre-determined list shown in the table). Separate figures are provided for males and females. For both males and females, the ADHD participants had more mixed sexual preferences than the Other participants, with medium effect sizes. Group membership explained 17.6% and 13.9% of the variance in sexual preference for females and males, respectively (i.e., medium effect size).

#### 3.2.2. History of Sexual Engagement, Sexual Affairs, and Satisfaction with Current Partner

Participants were asked about their relationship with their current partner (for those who had one). Table 4 shows that at the time of the study, significantly fewer participants in the ADHD group were in a relationship, sexual or otherwise (small effect size). For those currently in a relationship, both the ADHD and the Other groups considered this to be a monogamous relationship. There was no significant difference between the two groups in the frequency of monthly sexual activity with a current partner. The ADHD subjects reported significantly less satisfaction in their current relationships, both generally and more specifically with sexual intimacy (small effect size).

There was a significant relationship between current relationship satisfaction and sexual intimacy satisfaction for the whole sample (r = 0.472, *n* = 987, *p* < 0.001). The correlation was similar for females and males, 0.468 and 0.493, respectively. However, the correlation was significantly lower (z = 2.10, *p* < 0.05) in the ADHD group (r = 0.399, *n* = 364, *p* < 0.001) than the non-ADHD group (r = 0.509, *n* = 623, *p* < 0.001), accounting for 15.9% and 25.9% of the respective associations. 

A hierarchical regression model was used to investigate the extent to which current relationship satisfaction predicted sexual satisfaction after adjusting for age, sex, ADHD symptoms, anxiety, depression and currently taking ADHD medication (Step 1). Relationship satisfaction was added in Step 2. The results are provided for the final model in Table 5. The predictors in Step 1 accounted for 8.9% of the variance in sexual satisfaction. Relationship satisfaction, added in Step 2, accounted independently for 17.9% of the variance in sexual satisfaction. Anxiety diagnosis and ADHD medication did not contribute significantly to the final model.

### 3.3. Part 3: Sexual History

#### 3.3.1. Sexual Behavior

This section focused on the age of the participant during their first sexual relationship, the number of lifetime sexual partners, extra-partner sexual affairs, and sexual intercourse without use of contraception. Table 6 shows the results.

Female participants with a history of ADHD had sex at a significantly younger age (16.3 years) than Other female participants (17.1 years), but the difference was less than one year. No significant difference was found between the two groups for males. 

As far as the number of sexual partners are concerned, the distribution of each group of participants was highly skewed with a large range, which was 0 to 400 (median = 10) for females and 1 to 1000 (median = 9) for males. Fewer than 4% of the participants in both groups reported never having had a sexual relationship (age range = 18–58, median 26). Of those who had had a sexual relationship, for females, the lifetime median for sexual partners was significantly higher than that of the Other participants (12 and 8, respectively). For males, there was no significant difference in the lifetime number of sexual partners, the median being nine. 

Table 6 shows that there was a significant group difference in lifetime extra-partner sexual activities (infidelity) among females (medium effect size), but not among males. This measure excluded extra-partner affairs in the current relationship, and when these were specifically tested using a different question in the survey, the findings paralleled those found in Table 6. That is, females who reported a diagnosis of ADHD were significantly more likely than Other participants to report an extra-partner affair in their current relationships (χ^2^ = 8.18, df = 1, *p* < 0.01, Phi = 0.113). No significant differences emerged for males. 

Participants were asked about the frequency they had sexual intercourse without contraception (unless intentionally trying to have a baby). The six categories (Never, Rarely, Sometimes, Often, Very Often, and Always) were combined into two groups (Never–Sometimes vs. Often–Always, reflecting low risk and high risk, respectively).

For the 1256 participants who answered the question regarding the use of contraceptives, 377 (30.0%) fell in the high-risk group. There was an overall significant difference in high-risk-taking between males and females: 39.2% of the males and 25.2% of the females were classified as high risk-takers (χ^2^ = 25.9, df = 1, *p* < 0.001, Phi = 0.144). As far as ADHD for the total sample is concerned, there was no significant overall difference between the ADHD (31.2%) and Other participants (29.2%).

However, in a separate analysis for females and males with ADHD, as shown in Table 6, there was a significant difference in high risk-taking between females with ADHD and female Other participants (χ^2^ = 6.48, df = 1, *p* < 0.05, Phi = 0.089). No significant difference was found for males with or without ADHD. This suggests that only females with ADHD were at increased risk of regularly failing to use a contraceptive during sexual intercourse (i.e., being at increased risk of a sexually transmitted infection and/or pregnancy).

#### 3.3.2. Motivation behind Extra-Partner Sexual Relationship

We enquired about the motivation for infidelity from those participants who had owned purposefully engaging in sexual activities outside of their current or past romantic relationships. Options were selected from a predetermined list of 20 items (see Table 7). 

Table 7 shows that the ADHD participants claimed significantly higher scores than the Other participants on 11 (55%) of the 20 items. The largest differences were in response to poor impulse-control, alcohol disinhibition, sensation-seeking and feeling misunderstood, representing 23.0%, 16.2%, 14.6%, and 11.1% of the variance in the four motivation factors, respectively. These are all medium effect sizes.

We used discriminant function analysis to determine which of the variables in Table 7 best discriminated between the ADHD and Other participants. Overall, the model correctly classified 64.5% of the original groups’ cases. The structure matrix function loading of 0.400 or higher was as follows: Poor impulse-control (0.832), alcohol disinhibition (0.604), sensation-seeking (0.542) and feeling misunderstood (0.425).

#### 3.3.3. Sexually Transmitted Infections

Out of 1392 participants, 296 (21.3%) reported a history of one or more STIs, the most common being chlamydia (9.5%), followed by ‘genital/anal warts’ (6.6%) and ‘genital herpes’ (5.2%).

In the ADHD group, 132 (24.4%) reported one or more STI, in contrast to 164 (19.3%) among the Other participants. This difference was significant (χ^2^ = 4.1, df = 1, *p* < 0.05, Phi = 0.061]. The proportions were similar for females and males.

### 3.4. Part 4: Sexual Interests and Practices

#### 3.4.1. Pornography and Masturbation

Out of 1138 participants who responded to the question ‘Do you enjoy watching pornography?’, 317 (68.2%) of the ADHD and 477 (66.4%) of the Other participants said they enjoyed watching pornography. This difference is not significant, nor was there any significant difference in the frequency they watch it.

Most individuals watched pornography on their own, 334 (92.2%) of the ADHD and 499 (89.7%) of the Other group, and there was no significant difference between the two groups.

On a four-point scale (Never, Sometimes, Often and Always), participants with ADHD reported to masturbate significantly more often than the Other group when watching pornography (χ^2^ = 11.6, df = 3, *p* < 0.01, Cramer’s V = 0.112).

Hierarchical regression analysis was performed to ascertain if ADHD symptoms predicted the extent of masturbation to pornography after adjusting for age, sex and being in a current relationship. The data were entered in two steps. Step 1 (age, sex, current relationship), followed by Step 2 (adding ADHD). The results are shown in Table 8.

The four dependent variables explained 7.6% (low effect size) of the variance in frequency of masturbation; a younger age, having ADHD and being a male were the best predictors. In a separate analysis, adding the severity of the ADHD symptoms to the model in Step 2 did not add significantly to the variance in masturbation.

#### 3.4.2. Electronic Exchanges of a Sexual Nature

The participants were asked about sexual encounters on electronic devices, which included phone sex (‘phone sex for sexual gratification’), cybersex (‘online sex chat for sexual gratification’), sexting (‘exchanging sexually explicit messages’), sending and receiving sex images (both requested and unrequested) and intentionally seeking sexual encounters on dating apps. Out of the 1392 participants who responded, 273 (50.5%) of the ADHD and 363 (42.7%) of the Other participants claimed one or more of the eight items. The difference was significant (χ^2^ = 8.22, df = 1, *p* < 0.01; Phi = 0.076).

Significant differences between the two groups emerged. ADHD participants had a significantly greater number of exchanges than the Other participants: for phone sex (χ^2^ = 6.50, df = 1, *p* < 0.05; Phi = 0.070); cybersex (χ^2^ = 14.26, df = 1, *p* < 0.001; Phi = 0.104); sexting (χ^2^ = 7.16, df = 1, *p*< 0.01; Phi = 0.073); and intentionally seeking sexual encounters on dating apps (χ^2^ = 4.68, df = 1, *p* < 0.05; Phi = 0.059).

#### 3.4.3. Consensual Sexual Activities of Interest

The participants were asked ‘Which of the following consensual sexual activities interest you to the point that you would like to experience them (check all that apply)’. Table 9 provides the results. There were significant differences between the two groups for 11 (68.8%) of the items. The ADHD group had broader sexual interests than the Other group and a significantly higher overall score. The most impactful items in terms of effect sizes (all medium size) were ‘submissive role play’, ‘bondage’, ‘sex with a stranger’, ‘attending sex clubs and parties’ and ‘dominant role play’, representing 13.8%, 11.4%, 10.1%, 9.7%, and 9.0% of the variance, respectively.

#### 3.4.4. Consensual Sexual Activities Practiced

The participants were asked ’Have you ever engaged in the following consensual sexual activities (check all that apply)?’. Table 10 provides the results. The findings parallel those in Table 9 (sexual interests), although some of the effect sizes are lower. The dominant theme among the ADHD participants was more adventurous sexual practices. There were particular preferences for ‘submissive role play’, ‘dominant role play’ and ‘bondage’ (all medium effect sizes), representing 12.9%, 10.5% and 13.0% of the variance, respectively.

#### 3.4.5. Differences between Females and Males in Sexual Interests and Sexual Practice

There was a significant linear relationship between the total mean sexual interest and practice scores both for females (r = 0.653, *n* = 907, *p* < *0*.001) and males (r = 0.683, *n* = 485, *p* < 0.001). The corresponding correlations for ADHD and Other participants were 0.695 and 0.619 (*p* < 0.001), respectively. The size of the correlation was significantly larger (z = 3.47, *p* < 0.001) for females with ADHD (r = −0.717, *n* = 376, *p* < 0.001) than for Other female participants (r = 0.583, *n* = 531, *p* < 0.001). In contrast, the correlations for males with ADHD (r = 0.689, *n* = 165, *p* < 0.001) compared to the Other male participants (r = 0.675, *n* = 320, *p* < 0.001) were very similar. The findings suggest a stronger association between the sexual interest and practice scores among females than males with ADHD. This suggests that females with ADHD are relatively more likely than males with ADHD to practice what they would like to do in terms of adventurous sexual activities.

Table 9 and Table 10 show broad similarities in sexual interest and practice, although a further analysis revealed there were some robust sex differences. Table 11 shows the results of a discriminant functional analysis and separate analyses conducted for females and males for discriminating between the ADHD and Other participants. Structure matrices’ loadings show the similarities and differences between females and males with ADHD.

The main similarities between females and males in the structure matrix loadings were the consistently greater interest and more common practice among the ADHD participants in submissive role play, dominant role play and bondage. Both females and males in the ADHD group had greater interest than Other participants in group sex. In addition, ADHD females showed both greater interest and a greater likelihood of experiencing having sex with a stranger and threesomes. No such group differences were found for males. Sex with a stranger, group sex and open sexual relationships were confined to interest but not practiced so much.

## 4. Discussion

### 4.1. Clinical Conditions

Out of the 1392 participants, 864 (62.1%) claimed one or more clinical condition, which exceeds that of worldwide general population [20]. We deliberately targeted the recruitment of participants with ADHD via various ADHD patient support services; hence, our sample was selective in nature; 38.9% of the total sample reported a diagnosis of ADHD. A high number of comorbidities is commonly reported in adults with ADHD [21], and the most commonly reported conditions in the present sample were anxiety (29.0%) and depression (30%), together or alone, occurring in over one-third of the sample.

Consistently with previous studies [22], there was a high rate of co-occurrence between ADHD and autism spectrum disorder (ASD); over two-thirds of the ASD group also reported an ADHD diagnosis. For this reason, a separate analysis of those with ASD was not conducted, and we excluded participants reporting both conditions. This presents a limitation, since no direct inferences can be drawn specifically about the sexual interests, practices and relationships of adults with ASD.

Of the ADHD-diagnosed group, over half reported being currently prescribed ADHD medication and taking it regularly. A further 13.1% reported being prescribed ADHD medication but not taking it daily. Regular use of ADHD medication was not associated with the severity of ADHD symptoms. The most likely reason for this is that medication moderates symptom severity [23].

### 4.2. Sexual Orientation, Interests and Practices

Consistently with the findings of Hertz et al. [8], compared with the Other group, those in the ADHD group were substantially more adventurous in their outlook, interests and sexual practices. This was reflected in broader preferences for sexual partners, which included significantly greater engagement with partners of the same sex or both sexes. 

Similar findings have been reported for people with ASD, which also included non-binary participants [24]. In the present study 4.4% of participants classified themselves as non-binary; there is growing interest in the non-binary sexual orientation, and a separate paper will focus specifically on their sexual identity and its relationship with ADHD and comorbid problems. For the purpose of this manuscript, male and female correspond to a gender binary where the participant’s gender identity and natal sex align. In this context, female/male sex difference were used as mediating/moderating variables in the analyses.

The current findings corroborate another online survey that reported females with ADHD engaged in consensual sexual intercourse at a younger age, yet there was no significant difference for males [8]. In both studies, the effect size was low, and the age difference between ADHD and control groups, irrespective of participants’ sex, tends to be low, typically around one year, as found by the present study [4,6,8,25].

In line with previous research, the current survey found that more of the ADHD than Other participants had engaged in risky behavior. The former had a greater number of sexual partners; were more likely to participate in infidelity and unsafe sex without use of contraception; and had more sexual infections [6,8,9,10]. The effect sizes ranged from low to medium and were consistently stronger among the female than male participants. This tangible sex difference may reflect the large number of female participants in the current survey, providing increased statistical power. Treatment with ADHD medication may be helpful in mitigating some of this risk. A longitudinal study compared Taiwanese adolescents and young adults with ADHD (*n* = 17,898) with age and sex-matched controls (*n* = 71,592). They found 30% and 41% reductions in STIs with short-term and long-term use of medication, respectively [14]. Applying a similar methodology and when solely following up on a Taiwanese adolescent sample (ADHD *n* = 7500) and controls (*n* = 3020), Hua and colleagues [26] found that long-term use of ADHD medication was associated with a 30% reduction in teenage pregnancy. 

With respect to sexual interests and practices, we found a close overall linear relationship between the two, accounting for between one third and half of the joint variance. Interestingly, the strongest relationship was among females with ADHD, accounting for 51.1% of the shared variance, in contrast to 34.0% of females without an ADHD diagnosis. Consistent with this finding, the structure matrix loadings in the discriminant function analysis between ADHD and Other participants showed a similar pattern across interests and practices to that found among the male participants. 

Despite the robust sex difference, both female and male ADHD participants reported significantly more adventurous sexual interests and sexual practices than the Other participants. Females reported to have experienced their sexual interests (bondage, submissive role play and dominant role play—all medium effect sizes). By contrast, males reported specific interests but were less likely to engage in them. This suggests that females with ADHD are more likely to act out their sexual interests than males. This may reflect that females are more likely to act impulsively on interests and ideations; they may engage in promiscuous behavior in an attempt to gain friendships and/or social capital and in personal relationships may comply with requests that they may normally reject [27]. Psychoeducation and support for girls and women with ADHD is an ongoing need, and continuing the discussion around sexual behavior and health is an important, if sometimes neglected, component of this care. Targeting sexual health clinics with psychoeducation around the prevalence and experience of girls and women with ADHD is an intervention that could potentially lead to improvement in their health and wellbeing [28]. 

### 4.3. Relationship and Sexual Satisfaction

Compared with Other participants, there was a significantly lower association between sexual satisfaction and relationship satisfaction among the ADHD participants, and lower sexual and relationship satisfaction overall. Bijlenga et al. [12] reported similar findings for both males and females: they were less satisfied with their sex lives compared with controls. One possible explanation may be greater general reward deficiency in people with ADHD [29]. 

We enquired about the motivations for infidelity for participants who had admitted to purposefully engaging in sexual activities outside of their current or past romantic relationships. The most common reasons were poor impulse-control, alcohol disinhibition, sensation-seeking behaviors and feeling misunderstood (medium effect size). These four motivational factors are best described as increased ‘risk factors’ for infidelity (i.e., increased likelihood of it occurring) among people with ADHD. Whilst poor impulse-control is a key diagnostic feature of ADHD symptomatology [1], alcohol disinhibition, sensation-seeking and feeling misunderstood are best construed as exacerbating associated risk factors [30,31].

### 4.4. Strengths and Limitations

The main strengths of the current survey were the large samples size, the large number of female participants and the focus on the relationship between current sexual satisfaction and relationship satisfaction after controlling for the effects of age, natal sex and severity of ADHD symptoms.

The main limitation is the self-reported nature of the information provided, which could not be corroborated, including the diagnostic data and the selective nature of the participants. In addition, surveys, whether administered in written form or online, lack the flexibility and potential depth of information that can be elicited from face-to-face interviews. In future research, it might be helpful to ask about other relevant mental health conditions, such as eating and sleeping disorders, and prescribed medication, apart from those specific to ADHD. In addition, the current study was cross-sectional, testing participants only at one time-point and just following the international COVID-19 pandemic and the serious restrictions on social opportunities and behavior. Our survey specifically addressed this by stating, ‘Some of the questions ask about behavior “in a typical month”, these questions refer to experiences that occurred before the COVID-19 pandemic when social distancing measures were not in force.’ However, we acknowledge that this may have affected memory recall.

It is unclear whether the specific female difference found in the current and Hertz et al. [8] studies was influenced by the nature of the populations studied, which were online community samples rather than clinically referred studies and older participants typically participating in previous studies. Further research is needed to explain specific gender—including non-binary/gender diverse—differences in risky sexual behavior among people with ADHD, using both community and clinically referred participants. 

Another limitation is the general measures of relationship satisfaction and sexual satisfaction, which only provide an overall level of satisfaction rather than the range of different components that are likely to reflect the overall scores. For example, we did not focus specifically on sexual dysfunction, which has been found to be more prevalent among people with ADHD diagnosis and symptomatology [9,10].

## 5. Conclusions

The findings suggest that both sexes engage in risky sexual behaviors. However, the risk appears to be substantially greater for females with ADHD. Thus, these findings underscore the need for greater recognition and support for females who present with ADHD symptoms clinically. Using a harm-reduction model, early assessment and intervention of ADHD symptoms may extenuate the noted risks associated with impulse control and psychosexuality. Protective factors such as the education system may also play an important role through regular psychometric screening of ADHD symptoms in youth. As with all early recognition programs, the process that follows must allow for accessible psychological evaluations and responsive treatment by qualified health professionals. The development of an interdisciplinary harm reduction program may also benefit youth with ADHD symptoms through tiered interventions and psychosocial peer support.

In likeness to our recommendation for earlier intervention, similar harm-reduction principles can be applied within the scope of psychosexual health and safety. As poor impulse control is reportedly the most significant factor as to whether ADHD respondents engage in risky sexual behaviors, having readily accessible contraceptives can promote safety and sexual health. What is challenging about managing ADHD is that the problem may not occur due to a lack of self-discipline, but rather, as a means to use executive functioning abilities. Individuals with ADHD who have adequate self-awareness and self-regulation skills will still struggle to overcome executive functioning deficits compared to their neurotypical peers. For this reason, it may be helpful to think of ADHD as a disorder of performance; one may have the knowledge and skills to govern themselves, but a limited capacity to follow through consistently. It is imperative in a clinical setting to engage patients in an ongoing dialogue; including sexuality and sexual activity in an open manner may help to reduce the risk of negative outcomes.

## Figures and Tables

**Table 1 ijerph-20-02037-t001:** Clinical conditions claimed from a pre-determined list (*n* = 1392).

Clinical Condition	Total Sample *n* (%)	ADHD *n* (%)	Others *n*(%)
ADHD	541 (38.9)		
Depression or Major Depressive Disorder	417 (30.0)	211 (39.0)	206 (24.2)
Anxiety Disorder	404 (29.0)	211 (39.0)	193 (22.7)
Post-traumatic Stress Disorder	111 (8.0)	65 (12.0)	46 (5.4)
Specific Learning Disability	96 (6.9)	73 (13.5)	23 (2.7)
Obsessive Compulsive Disorder	59 (4.2)	39 (7.2)	20 (2.4)
Personality Disorder	47 (3.4)	26 (4.8)	21 (2.5)
Autism Spectrum Disorder	47 (3.4)	33 (6.1)	14 (1.6)
Bipolar Disorder	35 (2.5)	18 (3.3)	17 (2.0)
Asperger’s Syndrome	32 (2.3)	22 (4.1)	10 (1.2)
Substance Use Disorder	18 (1.3)	14 (2.6)	4 (0.5)
Traumatic Brain Injury	15 (1.1)	5 (0.9)	10 (1.2)
Oppositional Defiant Disorder	12 (0.9)	11 (2.0)	1 (0.1)
Psychotic Disorder	11 (0.8)	4 (0.7)	7 (0.8)
Social Communication Disorder	5 (0.4)	1 (0.2)	4 (0.5)
Conduct Disorder	5 (0.4)	2 (0.4)	3 (0.4)
Disruptive Mood Dysregulation Disorder	5 (0.4)	4 (0.7)	1 (0.1)
Intellectual Disability	1 (0.1)	0 (0.0)	1 (0.1)

**Table 2 ijerph-20-02037-t002:** Summary of the final hierarchical regression analysis model for variables predicting severity of ADHD symptoms as measured by the ASRS (*n* = 1324).

Explanatory Variable	B	SE B	Exp (B)
Age	−0.053	0.009	−0.125 **
Sex	−1.14	0.246	−0.101 **
Anxiety/depression/PTSD	1.786	0.241	0.165 **
ADHD diagnosis	6.168	0.298	0.565 **
ADHD Medication	0.735	0.356	0.056 *

Note. R^2^ = 0.394; ΔR^2^ = 0.122 for Step 1 (*p* < 0.001), 0.270 for Step 2 (*p* < 0.001), and 0.002 for Step 3 (*p* < 0.05). * *p* < 0.05; ** *p* < 0.001.

**Table 3 ijerph-20-02037-t003:** Differences in sexual preferences between ADHD and Other participants (*n* = 1371).

Sexual Preferences	ADHD [Females]	Others [Females]	χ^2^ (df = 2)	Cramer’s V
Females	24 (6.5%)	17 (3.3%)	27.5 **	0.176
Males	219 (59.0%)	391 (75.5%)		
Any/both genders	128 (34.5%)	110 (21.2%)		
Total	371(100%)	518 (100%)		
**Sexual Preferences**	**ADHD [Males]**	**Others [Males]**	χ **^2^ (df = 2)**	**Cramer’s V**
Females	111 (67.3%)	253 (79.8%)	9.37 *	0.139
Males	33 (20.0%)	37 (11.7%)		
Any/both genders	21 (12.7%)	27 (8.5%)		
Total	165 (100.0%)	317 (100.0%)		

* *p* < 0.01. ** *p* < 0.001.

**Table 4 ijerph-20-02037-t004:** Sexual engagement and relationship satisfaction for ADHD and Other participants.

Current Partner	ADHD *n* (%)	Others *n* (%)	Significance Tests
Currently has a partner	365 (71.1%)	625 (77.6%)	χ^2^ (df = 1) = 7.1 * (Phi = 0.073)
Relationship monogamous	331 (91.2%)	571 (91.5%)	χ^2^ (df = 1) = 0.03, ns.
Frequency of sexual activity per typical month (Q18):			
None 1 to 4 5 to 9 10 to 14 >15	62 (17.0%) 148 (40.7%) 91 (25.0%) 35 (9.6%) 28 (7.7%)	84 (13.5%) 225 (36.1%) 188 (30.1%) 80 (12.8%) 47 (7.5%)	χ^2^ (df = 4) = 7.45, ns.
Been unfaithful in current relationship	83 (23.2%)	115 (18.6%)	χ^2^ (df = 1) = 2.93, ns.
	**ADHD** **Mean (SD)**	**Others** **Mean (SD)**	**Significance Tests**
Relationship satisfaction	2.85 (1.39) [*n* = 365]	3.19 (1.18) [*n* = 625]	t = 4.05 (df = 988) ** (Cohen’s d = 0.267)
Sexual intimacy satisfaction	2.06 (1.54) [*n* = 366]	2.50 (1.44) [*n* = 628]	t = 4.57 (df = 992) ** (Cohen’s d = 0.301)

* *p* < 0.05; ** *p* < 0.001.

**Table 5 ijerph-20-02037-t005:** Summary of the final hierarchical regression analysis model for variables predicting sexual satisfaction (*n* = 963).

Explanatory Variable	B	SE B	Exp (B)
Age	−0.016	0.004	−0.132 ***
Sex	−289	0.090	−0.092 **
ADHD symptoms	−0.029	0.009	−0.103 **
Anxiety	−0.067	0.107	−0.020
Depression	−0.260	0.105	−0.078 *
On ADHD medication	0.068	0.107	0.019
Relationship satisfaction	0.506	0.033	0.432 ***

Note. R^2^ = 0.268; ΔR^2^ = 0.089 for Step 1 (*p* < 0.001) and 0.179 for Step 2 (*p* < 0.001). * *p* < 0.05; ** *p* < 0.01; *** *p* < 0.001.

**Table 6 ijerph-20-02037-t006:** Differences in riskier sexual history between ADHD and Other participants.

Life-Time History	ADHD	Others	Significance Tests
Age at first consensual sexual experience (Females)	16.33 (3.35) [*n* =337]	17.10 (3.07) [*n* = 454]	t = 3.36 (df = 789) ** [Cohen’s d = 0.383]
Age at first consensual sexual experience (Males)	17.65 (3.66) [*n* = 274]	17.22 (2.95) [*n* = 139]	t = 1.22 (ns)
Life-time number of sexual partners (Females)	Median = 12 [*n* = 322]	Median = 8 [*n* = 464]	U = 87,780.00 Z = 4.18 ** [r = 0.149]
Life-time number of sexual partners (Males)	Median = 9.5 [*n* = 140]	Median = 9 [*n* = 274]	U = 18,580.00 Z = −0.402
Extra-partner sexual activities (Females)	179 (52.8%)	189 (41.2%)	χ^2^ (df = 1) = 10.61 ** [Phi = 0.115]
Extra-partner sexual activities (Males)	68 (48.9%)	129 (45.9%)	χ^2^ (df = 1) = 0.339
Sex without contraception (Females):			
Never-Sometimes Often-Always Total	243 (70.2%) 103 (29.8%) 346 (100%)	373 (78.0% 125 (22.0%) 478 (100%)	χ^2^ (df = 1) = 6.48 * [Phi = 0.089]
Sex without contraception (Males):			
Never-Sometimes Often-Always Total	94 (65.3%) 50 (34.7%) 144 (100%)	169 (58.7%) 119 (41.3%) 288 (100%)	χ^2^ (df = 1) = 1.75

* *p* < 0.05; ** *p* < 0.001.

**Table 7 ijerph-20-02037-t007:** Differences between ADHD and other participants in the motivation behind extra-partner affairs from a pre-determined list (*n* = 826).

	ADHD (*n* = 335)	Others (*n* = 491)	χ^2^ (df = 1)	Phi
Opportunity	92 (28.5%)	115 (19.7%)	1.73	
Sensation-seeking	157 (46.9%)	159 (32.4)	17.68 ***	0.146
Developed via social media/online relationship	41 (12.2%)	53 (10.8)	0.414	
Re-contact with an ‘old flame’.	72 (21.5%)	79 (16.1%)	3.89 *	0.049
Aroused by watching pornography	19 (5.2%)	13 (2.6%)	4.89 *	0.027
Poor impulse-control	112 (33.4%)	69 (14.0%)	43.71 ***	0.230
Alcohol disinhibition	120 (35.8%)	104 (21.2%)	21.59 ***	0.162
Drug use disinhibition	38 (32.9%)	32 (28.2%)	5.93 *	0.085
Peer pressure	5 (1.5%)	8 (1.6%)	0.024	
Peer pressure from another person	15 (4.5%)	19 (3.9%)	0.186	
Feeling bored	59 (17.6%)	55 (11.2%)	6.88 **	0.091
Feeling unhappy	116 (34.6%)	128 (26.1%)	7.00 **	0.092
Feeling unappreciated	99 (29.6%)	111 (22.6%)	5.07 *	0.078
Feeling misunderstood	58 (17.3%)	48 (9.8%)	10.11 **	0.111
Feeling angry	29 (8.7%)	34 (6.9%)	0.85	
Revenge	25 (7.5%)	23 (4.7%)	2.81	
Incompatible libido with usual partner	60 (17.9%)	74 (15.1%)	1.18	
Usual partner couldn’t have sex	12 (3.6%)	20 (4.1%)	0.129	
Seeking unusual sexual activity	20 (6.9%)	35 (7.1%)	0.430	
Other (unspecified)	51 (15.2%)	40 (8.1%)	10.17 ***	0.111

* *p* < 0.05; ** *p* < 0.01; *** *p* < 0.001.

**Table 8 ijerph-20-02037-t008:** Summary of final hierarchical regression analysis model for variables predicting extent of masturbation to pornography.

Explanatory Variable	B	SE B	Exp (B)
Age	−0.020	0.003	−0.249 ***
Sex	0.175	0.066	0.087 **
In a relationship	−0.152	0.074	0.066 *
ADHD diagnosis	0.244	0.066	0.119 ***

Note. R^2^ = 0.076; ΔR^2^ = 0.014 for Step 2 (*p* < 0.001). * *p* < 0.05; ** *p* < 0.01; *** *p* < 0.001.

**Table 9 ijerph-20-02037-t009:** Differences in sexual interests between ADHD and Other participants (*n* = 1392).

	ADHD	Others	χ^2^ (df = 1)	Phi
Sex with a stranger	154 (28.5%)	168 (19.7%)	14.2 ***	0.101
Threesome	210 (38.8%)	273 (32.1)	6.6 *	0.069
Group sex (>three people)	113 (20.9%)	120 (14.1%)	10.9 ***	0.089
Open sexual relationships	103 (19.1%)	109 (12.8%)	10.4 **	0.085
Swinging with other partners	82 (15.2%)	103 (12.1%)	2.7	
Attending sex clubs/parties	116 (21.4%)	119 (14.0%)	13.1 ***	0.097
Oral sex	306 (56.6%)	449 (52.8%)	1.9	
Anal sex	178 (32.9%)	240 (28.2%)	3.5	
Submissive role play	208 (38.5%)	217 (25.5%)	29.4 ***	0.138
Dominant role play	144 (26.7%)	162 (19.0%)	11.2 ***	0.090
Sex toys	280 (51.8%)	400 (47.0%)	3.0	
Spanking	153 (28.3%)	212 (24.9%)	1.9	
Bondage	167 (30.9%)	177 (20.8%)	18.0 ***	0.114
Voyeurism	109 (20.1%)	121 (14.2%)	8.4 **	0.078
Autoerotic asphyxiation	45 (8.3%)	44 (5.2)	5.5 *	0.063
Mean total score	4.42 (SD = 4.1)	3.45 (SD = 3.7)	t = −4.54 ***	Cohen’s d = −0.250

* *p* < 0.05; ** *p* < 0.01; *** *p* < 0.001.

**Table 10 ijerph-20-02037-t010:** Differences in sexual practices between ADHD and Other participants (*n* = 1392).

	ADHD	Others	χ^2^ (df = 1)	Phi
Sex with a stranger	142 (26.4%)	165 (19.4%)	14.2 **	0.081
Threesome	131 (24.2%)	170 (20.0%)	3.5	
Group sex (>three people)	53 (9.8%%)	56 (6.6%)	4.7 *	0.058
Open sexual relationships	46 (8.5%)	49 (5.8%)	3.9 *	0.053
Swinging with other partners	24 (4.4%)	25 (2.9%)	2.2	
Attending sex clubs/parties	49 (9.1%)	51 (6.0%)	4.7 *	0.058
Oral sex	402 (74.3%)	596 (70.0%)	2.9	
Anal sex	248 (45.8%)	334 (39.2%)	5.9 *	0.065
Submissive role play	169 (31.2%)	169 (19.9%	23.3 **	0.129
Dominant role play	136 (25.1%)	141 (16.6%)	15.2 **	0.105
Sex toys	300 (55.5%)	413 (48.5%)	6.3 *	0.067
Spanking	180 (33.3%)	278 (32.7%)	0.05	
Bondage	156 (28.8%)	151 (17.7%)	23.7 **	0.130
Voyeurism	50 (9.2%)	53 (6.2%)	4.4 *	0.056
Autoerotic asphyxiation	44 (8.1%)	49 (5.8%)	3.90	
Mean total score	3.97 (SD = 3.4)	3.19 (SD = 3.1)	t = −4.39 **	Cohen’s d = −0.241

* *p* < 0.05; ** *p* < 0.001.

**Table 11 ijerph-20-02037-t011:** Discriminant functional analysis (structure matrix) of the differences between sexual interests and practices for ADHD and Other participants (*n* = 1392).

	Females		Males	
	Interests (*n* = 907)	Practices (*n* = 907)	Interests (*n* = 485)	Practices (*n* = 485)
Sex with a stranger	**0.422**	**0.478**	**0.702**	0.262
Threesome	**0.455**	**0.414**	0.245	−0.068
Group sex (>three people)	**0.465**	0.293	**0.412**	0.302
Open sexual relationships	0.367	0.313	**0.551**	0.181
Swinging with other partners	0.185	0.304	0.342	−0.047
Attending sex clubs/parties	**0.510**	0.332	0.412	0.248
Oral sex	0.259	0.398	0.131	−0.166
Anal sex	0.334	0.378	0.297	0.163
Submissive role play	**0.673**	**0.558**	**0.403**	**0.484**
Dominant role play	**0.441**	**0.535**	**0.460**	**0.408**
Sex toys	0.277	0.366	0.052	0.084
Spanking	0.072	−0.041	0.332	0.115
Bondage	**0.453**	**0.515**	**0.597**	**0.646**
Voyeurism	**0.431**	0.226	0.350	0.374
Autoerotic asphyxiation	0.237	0.206	0.271	0.117
Mean score	44.51 *	45.98 *	23.19	23.15

Loadings of 0.40 or above are highlighted in bold. * *p* < 0.05.

## Data Availability

The survey, questionnaire and data can be obtained by contacting the lead author.

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
