# Peer review of "Let’s Talk about Sex… and ADHD: Findings from an Anonymous Online Survey"

_ijerph, 2023, doi:10.3390/ijerph20032037_

Round 1
Reviewer 1 Report
This is a very long paper. The authors need to be more selective about which data to include. They should also be more concise and to avoid repetition. I recommend that the authors rewrite the paper with the aim of reducing the word count by at least 50%.
The authors recruited 1466 respondents to an online survey.
The 34 item survey was not available to review in the additional files submitted. It appears that in the questions about preferences and behaviour patterns, the participants were asked to estimate eg in frequency of use of contraception,
It is somewhat confusing about which participants were excluded. This needs to be clearly illustrated in a flow diagram, that gives the number of non-binary in the ADHD and non-ADHD groups and who were excluded; also the number of ASD participants excluded from the controls (‘too small for meaningful analysis’).
The description of the survey should be far more concise, avoiding unnecessary information. The authors have defined ‘romantic relationship’ but no definition is given for ‘cybersex’ and ‘sexting’.
Far too much statistical detail is given. For the analyses where there is no significant difference between groups (eg pornography), this should simply be stated and the full analysis should not be given.
The paper is poorly edited – eg % left off Table 9; missing bracket line 161; extra full stop line 233. Tables 2, 4, 5, 7 – define the * in the legend. Line 539 STIs not STI’s.
Line 299 – add other ‘female’ participants.
Table 1 – should give the number and % for the ADHD and non-ADHD separately, as is done in Table 3. Also include in this the demographics (age, sex) and the mean ASRS score.
Table 2 – in title ‘ADHD severity of symptoms’ – should state how this was calculated eg using ASRS.
Table 3 – Sexual preference – it seems that at most only 2/3 of the ADHD respondents were heterosexual, whereas in non-ADHD this is approaching 80%. It would be good to know if the wording of the question may have been confusing and to have some verification by cross-checking with the sex of the current partner.
In reporting general relationship satisfaction and sexual intimacy satisfaction it is important to be clear about whether this relates to relationship satisfaction in general or to the current relationship (eg line 278).
Line 302 – give the medians.
Line 303 – give the age range and median of the 4% who had never had sex.
Table 7 – state whether column 1 is a predetermined list. If so, why is ‘Re-contact with an ‘old male’’ not included?
The summary of the findings at the start of the discussion should be far more concise and report only the main findings.
Line 499 -add ‘either or both’
Shorten the sentence line 509-512 to: ‘The most likely reason for this is that medication moderates symptom severity [23].’
Lines 550-599 is repeating too much of the results.
Lines 624-632 appears to be unsupported opinion.
Points that could usefully be included in the discussion are that perhaps lower relationship/sexual satisfaction might relate to reward deficit – see Blum K et al. ie less satisfaction with everyday life leading to reward seeking behaviour, including sexual behaviour.
Regarding females being more likely to engage in sexually adventurous/risky behaviour, this might relate to females having more opportunity – if males are generally more promiscuous than females, there would be more willing male partners available for promiscuously-inclined females.
Author Response
Thank you for your review, it was helpful in improving the paper. We have uploaded a detailed response.

Reviewer 2 Report
Young et al. conducted an online anonymous survery of sexual satisfaction and behavioral in ADHD adults and reported their findings in detail. The relationship between sexuality and psychiatric disorders is a topic with increasing interest. This study fills the gaps in those with ADHD. The whole manuscript was well written and showed readers the most important results. There are some concerns could be considered.
-Regarding the diagnostic criteria of psychiatric disorders: considering the limitations of the information source of the online survey, whether the clinical diagnosis of disorders comes merely from self-report? Whether any relevant proofs are required? Those who remitted patients with one or more psychiatric disorders in past could also participate in this survey? Please specify this information in section of participants
-Did the survey asked only about medication for ADHD? What about for those participants with major depression, bipolare, schiyoprenia or OCD? For these disorders pharmacotherapy are also important. Moreover, there are date showing the these medications affect sexual function as side effect and probably affect indirectly life/sexual satisfaction. Could you please provide some description to this point?
-Sleep disorder and eating disorder are common comorbidities to ADHD? However, this is no information in Table 1. Could authors please describe briefly about this information or explain why these two disorders are not considered?
Minor concers:
-Line 19-20: relationships repeated.
-Line 48: what does „community controls“ mean?
-Line 218: please keep uniformly one demical place or two demical places when presenting descpiptive statstics. This applies to the full text.
-Line 220: one point too much
-Full text: the zero to the left of the demical point for effect size and statistical tests cannot be omitted. Please check for full text.
-Line 273: All but six (0.73%) had reported an ADHD diagnosis. In this sentence 0.73 is a wrong result here. It might also be caused by incompleted sentence, please correct this.
-Line 233: one point too much
-Line 240: Table 2. -144, point after the minus sign is missed, please correct this
-Line 242: Table 2. please indicate the p-value range represented by the asterisk
-Line 262 and Table 4: results of Chi square tests were different in text and in Table. Please check this.
-Line 276: Table 24. please indicate the p-value range represented by the asterisk
-Line 278: point for the p-value is missed
-Line 334: wrong use of asterisk of representing p-value range, please correct this.
-Line 350: Table 7. please indicate the p-value range represented by the asterisk
-Line 369: ADHD reported to masturbate significantly more the Other… word missing?
-Line 370: repeated degrees of freedom. Please correct this
-Line 381: please add semicolon between .05 and p and unify the location of asterisk
-Line 387-392: for 2x2 chi square tests the degress of freedom are alwalys 1, thus there is no neccessary to privode df. However, if you would like to do so, please unify their format
-Line 401-403: one proportion of explained variance was lack, please add this
-Line 414-415: please add p-values to the corresponding coefficient
-Line 416-417: please provide z score as you did in line 280 when you comparing the correlation coefficients or r-values
-Line 438: see above
-Line 440: see above
-Line 443: see above
Author Response
Thank you for your review, it was very helpful in improving the paper. We have uploaded a detailed response.

Round 2
Reviewer 2 Report
I have no further comments.